# Impact of Scene Content on High Resolution Video Quality

**DOI:** 10.3390/s21082872

**Published:** 2021-04-19

**Authors:** Miroslav Uhrina, Anna Holesova, Juraj Bienik, Lukas Sevcik

**Affiliations:** Department of Multimedia and Information-Communication Technology, University of Zilina, Univerzitna 1, 010 26 Zilina, Slovakia; anna.holesova@uniza.sk (A.H.); juraj.bienik@uniza.sk (J.B.); lukas.sevcik@uniza.sk (L.S.)

**Keywords:** ACR, H.264/AVC, H.265/HEVC, QoE, subjective assessment

## Abstract

This paper deals with the impact of content on the perceived video quality evaluated using the subjective Absolute Category Rating (ACR) method. The assessment was conducted on eight types of video sequences with diverse content obtained from the SJTU dataset. The sequences were encoded at 5 different constant bitrates in two widely video compression standards H.264/AVC and H.265/HEVC at Full HD and Ultra HD resolutions, which means 160 annotated video sequences were created. The length of Group of Pictures (GOP) was set to half the framerate value, as is typical for video intended for transmission over a noisy communication channel. The evaluation was performed in two laboratories: one situated at the University of Zilina, and the second at the VSB—Technical University in Ostrava. The results acquired in both laboratories reached/showed a high correlation. Notwithstanding the fact that the sequences with low Spatial Information (SI) and Temporal Information (TI) values reached better Mean Opinion Score (MOS) score than the sequences with higher SI and TI values, these two parameters are not sufficient for scene description, and this domain should be the subject of further research. The evaluation results led us to the conclusion that it is unnecessary to use the H.265/HEVC codec for compression of Full HD sequences and the compression efficiency of the H.265 codec by the Ultra HD resolution reaches the compression efficiency of both codecs by the Full HD resolution. This paper also includes the recommendations for minimum bitrate thresholds at which the video sequences at both resolutions retain good and fair subjectively perceived quality.

## 1. Introduction

In recent years, the number of various types of surveillance and data collection cameras located both indoors and outdoors have been constantly increasing. Popularity of home security cameras is also growing as even high-quality models become more affordable. Typical surveillance cameras applications include public safety, protection of facilities against theft or vandalism, remote video monitoring, traffic surveillance, weather monitoring, or more special cases, such as animal monitoring or data collection, for statistical or marketing purposes. Today, due to the pandemic situation, face recognition with and without a protective mask is also becoming a point of interest for researchers in cooperation with technology companies [1,2,3]. It is important to realize that each such employed sensor produces a tremendous amount of data to be subsequently transmitted over the network or further processed, which calls for effective video compression. Furthermore, whether the image or video is presented to a live person or a machine learning algorithm (most often for its classification or segmentation), the best results can be achieved when the image is of the highest achievable quality. This implies one common goal for the distributors, communication service providers, or even broadcasting companies, to optimally set the compression parameters so that perceived video quality is maximal, while the bandwidth requirements are minimal. This challenge leads to increased interest in the analysis of video content followed by the individual setting of the compression parameters of video sequences with different types of scene content. Even though many studies deal with the video quality assessment using subjective methods, the demand exceeds the supply; there is still a lack of video quality datasets, as well as recorded subjective tests, conducted on these datasets. Very popular and extensively used datasets, such as References [4,5,6,7,8,9,10,11,12,13,14,15,16,17,18,19,20,21,22,23,24,25,26,27], come from the University of Texas and were developed by the Laboratory for Image and Video Engineering. Another very popular option is the VQEG-HDTV database [28], which is a result of international project of VQEG (Video Quality Experts Group) consortium. Other well-known datasets are BVI-HD [29], BVI textures [30] and BVI-HFR [31] developed at the University of Bristol, AVT-VQDB-UHD-1 database [32] made by the Ilmenau University of Technology, Ultra Video Group (UVG) dataset [33] composed at the Tampere University, SJTU 4K video quality database [34] from the Shanghai Jiao Tong University, Image and Video Processing Subjective Quality Video Database [35] developed at the Chinese University of Hong Kong, collection of IRCCyN/IVP databases from the Institut de Recherche en Communications et Cybernétique de Nantes [36], Konstanz Natural Video Database (KoNViD-1k) made by the Universitaet Konstanz [37,38], MCL-V [39], and [40] databases from the MSC University of Southern California, Scalable Video Database [41,42] composed at the EPFL, ReTRiEVED Video Quality Database [43] made by the Universita Degli Studi or TUM databases [44,45] developed by the Technical University of Munich. Taking into account the demand and importance of measuring the performance of video quality assessment techniques, a number of studies on perceptual evaluation was presented. Most of them merely compare the quality of video sequences with various characteristics evaluated by different subjective methods and do not examine the content aspect. Rerabek et al. [46] examined a rate-distortion performance analysis and mutual comparison of one of the latest video coding standards H.265/HEVC with VP9 codec. Ramzan et al. [47] presented a performance evaluation of three coding standards—Advanced Video Coding (H.264/MPEG-AVC), High-Efficiency Video Coding (H.265/MPEG-HEVC), and VP9, based on subjective and objective quality evaluations. Two different sequences at both resolutions (Full HD and Ultra HD) were tested using the DSIS method. Bienik et al. [48] measured the impact of the compression formats, namely H.264, H.265, and VP9 on perceived video quality. The evaluation was performed on four Full HD sequences using the Absolute Category Rating (ACR) and DSCQS methods. Xu et al. [49] presented a subjective video quality assessment on 4K Ultra-High Definition (UHD) videos using the DSCQS method. Six different test sequences were used for the evaluation.Herrou et al. [50] focused on a performance comparison between HEVC and VP9 in the HDR context through both objective and subjective evaluations. Dumic et al. [51] offered findings on subjective assessment of H.265 versus H.264 Video Coding for High-Definition Video Systems. For the evaluation, a database consisting of 120 degraded HD video sequences with 4 contents encoded at various compression rates to H.265/HEVC and H.264/AVC formats was compiled. Milovanovic et al. [52] subjectively compared the coding efficiency of three video coding standards (MPEG-H HEVC, H.264/MPEG-4 AVC, and H.262/MPEG-2). Sotelo et al. [53] presented a subjective quality assessment of HEVC/H.265 compressed 4K Ultra-High-Definition (UHD) videos in a laboratory viewing environment. Kufa et al. [54] explored coding efficiency performance of High Efficiency Video Coding (HEVC) and VP9 compression formats on video content in Full HD and UHD resolutions. Deep et al. [55] focused on the comparison of HEVC and VP9 based on both subjective and objective evaluation on various (720p, 1080p, and 2160p) test videos. Akyazi et al. [56] examined the compression efficiency of HEVC/H.265, VP9, and AV1 codecs based on subjective quality assessment. Our survey of research papers shows that there is still a lack of databases of video sequences annotated according to subjective evaluation. Therefore, this paper brings new subjective results and also explores the impact of the video content on the subjective assessment. We decided to compare today’s most used compression standards—H.264/AVC and H.265/HEVC—on video sequences at Full HD and Ultra HD resolutions. Our publication follows Reference [57], where a new 4K video dataset was compiled with full subjective scores (Mean Opinion Score (MOS)) of videos at different bitrates compressed by HEVC/H.265 codec evaluated by the Double Stimulus Impairment Scale (DSIS) method, variant II. For our measurements, we decided to use the Absolute Category Rating (ACR) method.

## 2. Dataset Description and Preparation

### 2.1. Dataset Description

For our measurements, we used the dataset from the Media Lab of the Shanghai Jiao Tang University [34]. We selected eight sequences with various scene content, illustrated in Figure 1, classified according to the Temporal Information (TI) and Spatial Information (SI) from this database. SI defines the amount of spatial detail in an image and is higher for more spatially complex scenes, while TI represents the number of temporal changes in a video sequence and is higher for high motion sequences [58]. The spatial perceptual information is based on the Sobel filter and is represented by the formula: (1)$$\begin{array}{c} SI=max_{time}\left[std_{space}\left[Sobel\left(F_{n}\right)\right]\right], \end{array}$$
where $F_{n}$ stands for video frame, and $std_{space}$ for the standard deviation over the pixels in each Sobel-filtered frame. The temporal information is computed as:
(2)$$\begin{array}{c} TI=max_{time}\left[std_{space}\left[M_{n}\left(i,j\right)\right]\right], \end{array}$$
where $M_{n}\left(i,j\right)$ is the difference between pixels at the same position in the frame belonging to two consecutive frames, i.e.,
(3)$$\begin{array}{c} M_{n}\left(i,j\right)=F_{n}\left(i,j\right)-F_{n-1}\left(i,j\right), \end{array}$$
where $F_{n}\left(i,j\right)$ is the pixel at the *i*-th row, and *j*-th column of *n*-th frame in time [58]. Both of these parameters were calculated for each sequence using the Mitsu tool [59] and plotted in Figure 2. The general specification of the dataset is given in Table 1, and the content of individual sequences is briefly described in Table 2.

### 2.2. Dataset Preparation

In our research, we decided to explore the quality of 8-bit video sequences at two commonly used resolutions, i.e., Full HD (FHD) and Ultra HD (UHD) with a typical chroma subsampled YUV 4:2:0 format. Because original sequences were uncompressed and YUV 4:4:4 color format at Ultra HD resolution with 10-bit depth was used, we had to convert them to the appropriate formats. Therefore, all test sequences were first chroma subsampled from YUV 4:4:4 to YUV 4:2:0 format and also the bit depth was changed from 10 to 8 bits per channel. Subsequently, all these conversion steps were repeated for Full HD resolution utilizing the FFmpeg tool [61]. As we wanted to assess also Full HD in addition to Ultra HD, the resolution also had to be altered. For all these conversion steps we used once again the FFmpeg tool [61]. Correspondingly, two uncompressed test sequences were generated (Figure 3) for each type of content, which adds up to 16 videos. We call them the source video sequences (SRCs) for the rest of this paper.

### 2.3. Coding Process

All these test video sequences (SRCs) were afterwards encoded to both compression standards to be evaluated, i.e., H.264/AVC and H.265/HEVC. As a quality restriction parameter, we decided to use the constant bitrate. We selected 5 various target bitrates: 1, 3, 5, 10 and 15 Mbps based on our previous research [62] which have shown the efficiency of codecs growing nonlinearly with increasing bitrate. We have limited the number of bitrates to 5 as a compromise between the complexity and time requirements of subjective testing and precision of the measurements. For the purposes of our research, we decided to use the Group of Pictures (GOP) length typical for video intended for transfer over a noisy communication channel. The GOP length is based on the framerate of used video sequences and is commonly set to half of the framerate value. Accordingly, given that test video sequences had a framerate of 30 fps (frames per second), we chose the GOP length of 15 frames, i.e., M = 3, N = 15. The first number, labeled with M letter, expresses the distance between two anchor frames (I or P) and the second number, denoted with N letter, stands for the distance between two key frames (I). For this coding process, we used once again the FFmpeg tool, which contains libraries x264 and x265 for H.264/AVC and H.265/HEVC codec, respectively [61], creating the total of 160 video sequences for the subjective quality assessment. We refer to them as PVSs (Processed Video Sequences) for the rest of this paper. The FFmpeg command example for encoding the Wood test sequence to the H.264 format at 1 Mbps bitrate is: *ffmpeg -i Wood_1920x1080_30fps_420_8bit_YUV.yuv -vcodec libx264 -command example-params keyint=15:min-keyint=15:bframes=3:b-adapt=1:bitrate=1000:vbv-maxrate=1000:vbv-bufsize=1000 Wood_1920x1080_30fps_420_8bit_H264_01M.mp4.*

## 3. Subjective Quality Assessment

During the subjective testing, all created PVSs were shown to people of different ages and genders to evaluate their quality. We decided to use the Absolute Category Rating (ACR) method [58,63] which belongs to the category of Single Stimulus (SS) subjective video quality assessment techniques. The principle of this method is that the degraded sequences are presented to the observers one at a time, and they are asked to rate its quality on a five-level grading scale, where 1 indicates the bad quality, and 5 stands for the excellent quality. The measurement was conducted in two laboratories separately: one situated at the University of Zilina (UNIZA), and the second at the VŠB – Technical University in Ostrava. The video sequences were presented on three types of displays (Table 3) depending on the resolution of the test sequences in the laboratories under normal indoor illumination conditions.

Thirty participants, mostly students, were involved in the testing in each laboratory. All of them were naive observers which means they had no expertise in the image artefacts that may be introduced by the system under test. Naturally, they were thoroughly acquainted with the method of assessment, types of impairment, grading scale, sequence, and timing as required by Reference [58]. The statistical distribution of the number of men and women who took part in the tests, as well as the average age of all observers, is shown in Table 4. The course of the entire subjective assessment process is represented by Figure 4.

## 4. Statistical Analysis and Presentation of the Results

After performing the subjective tests, we processed all collected results statistically; for each test sequence, codec, and resolution, the Mean Opinion Score (MOS) and 95 percent Confidence Interval (CI) in accordance with Reference [64] were calculated and plotted in graphs, a shown below. The presentation of the results could be divided into five parts. In the first part, the cross-comparison of the results obtained from different laboratories, i.e., from UNIZA and VŠB, is performed using the Pearson correlation coefficient (PCC) and the Root Mean Square Error (RMSE). In the second part, the bitrate impact on the perceived video quality depending on the scene content is plotted. The third part deals with the Analysis of Variance (ANOVA) which was applied on the acquired data. In the fourth part, the impact of the bitrate on the perceived video quality in terms of the used codec and resolution is presented. Finally, in the fifth part, the minimum bitrate thresholds at which the video sequence should be encoded to reach certain quality are determined.

### 4.1. Correlation between the Results from Individual Laboratories

To compare the MOS values obtained from both laboratories, i.e., from UNIZA and VŠB, and, to find out the correlation, the Pearson correlation coefficient (PCC), as well the Root Mean Square Error (RMSE) were calculated. All computations were done for both codecs and resolutions, as well as for all test sequences. The results are plotted in Figure 5 and Figure 6 and are shown in Table 5.

As we can see from Figure 5 and Figure 6, as well as from Table 5, there is a high correlation between the results from both laboratories. The lowest correlation was reached by the combination of Full HD resolution and H.264 codec. This is most likely due to the different displays used in the assessments; at the UNIZA laboratory, the Full HD display was used, while, at the VŠB laboratory, the Ultra HD display was used. Vice versa, the highest correlation rate was achieved by video sequences encoded to H.264 at UHD resolution.

### 4.2. Impact of Bitrate on Video Quality Depending on Scene Content

Figure 7 shows the impact of the bitrate on the perceived video quality (defined by the MOS with associated CI). In this figure, eight graphs are inserted considering used codec, resolution, and laboratory where the evaluation was conducted. Sequences with different scene contents are color-coded in the graphs; each curve represents MOS values for a given test sequence. Figure 8 shows the average MOS values obtained from UNIZA and VSB laboratories.

It is apparent from the graphs that the sequences with the lowest SI and TI values, such as the “Bund Nightscape” and the “Construction Field”, reached the best MOS value. Vice versa, the observers rated the sequences situated in the middle of the SI-TI diagram, such as the “Marathon” or “Runners”, as of worst quality. Interesting cases are the “Campfire Party” and the “Fountains” sequences. The “Campfire Party” contains a lot of movement (high TI values) but not many details (low SI values) and reached low MOS value, while the “Fountains” sequence lies near to the “Bund Nightscape” and the “Construction Field” sequences, meaning it has low both TI and SI values and also scored low on the MOS scale. A special case is the “Wood” sequence which is situated at the upper right corner of the SI-TI diagram. Nevertheless, its quality was perceived as similar to the sequences “Fountains” and “Runners”. All these differences are more pronounced:at low bitrates—with increasing bitrate, the perceived quality rises, too, and approaches the perceived quality of sequences with low SI-TI values,at Ultra HD resolution rather than at Full HD resolution, andat H.265 codec rather than at H.264 codec.

Based on these results, we can state that the compression efficiency and related video quality depends on the content of the sequences. However, the sequence representation and description only by the spatial and temporal information is not sufficient and should be the subject of further research. We suggest other parameters should be used to describe the scene, such as, for instance, the luminance and contrast or the colors occurring in the scene. In addition, the psychological factors should be considered. Based on the results, we can also state that the temporal information has greater impact on the perceived quality than the number of the objects defined by the spatial information.

### 4.3. Analysis of Variance

To verify what stemmed from the graphical representation of the subjective evaluation results, the ANOVA was applied on the data [65]. The three-way ANOVA was used to compare the significance and influence of individual sequence parameters on the resulting perceived video quality. The interaction between three independent variables, bitrate (X1), content (scene type) (X2), and resolution (X3) in Table 6 or compression standard (X3) in Table 7 was examined, with video quality being considered a dependent variable. Table 6 and Table 7 depict the three-way ANOVA matrices. The *F*-value, also called the F-ratio is calculated as the variance of the group means divided by the mean of the within group variances (Mean Squared Error). Greater *F*-value indicates more significant variation. In ANOVA, the *p*-value, i.e., the probability of getting the observed result at random, is also determined. For the source of variation to be regarded as insignificant, the *p*-value must be higher than a given alpha level, commonly set to 0.05. When performing ANOVA, the *p*-value is also determined to investigate the probability of rejecting the hypothesis.

Based on the analysis of the tables, the following conclusions can be drawn. Table 6 indicates that for H.265 encoded sequences, the effect of resolution can be ignored, since this variable was deemed statistically insignificant. In contrast, in the case of the H.264 codec, this negative phenomenon does not occur and resolution is the second most important parameter that determines the subjectively perceived quality. For both codecs, an alteration in bitrate results in a maximum change in the subjective MOS. According to Table 7, the impact of compression format on the perceived quality is considered statistically insignificant for Full HD video sequences. However, that is not the case for Ultra HD resolution, where deployed codec is the second most influential variable. Equivalently to Table 6, the bitrate has the greatest effect on the subjective video quality assessment results. All remaining ANOVA test results in both tables can be regarded statistically significant based on their *p*-values.

### 4.4. Impact of Bitrate on Video Quality Depending on Codec and Resolution

Figure 9 shows the impact of the bitrate on the perceived video quality (defined by the MOS with associated CI) plotted separately for each type of video sequence. In this figure, eight graphs are inset, considering examined test sequence, which show the impact of used codec and resolution on the perceived quality of a given sequence; curve represents averaged MOS values from both laboratories for a given codec and resolution.

In Figure 10, the averaged MOS value from both laboratories from all used test sequences for each codec and resolution is plotted.

We can draw several conclusions from Figure 9 and Figure 10. Firstly, it is apparent that the H.265 compression standard yields better quality than the H.264 codec. This is a generally known fact and we expected it. But what is interesting and important is that the efficiency difference between these two codecs is negligible for the Full HD video sequences. Therefore, it is inessential to use H.265 compression standard at this resolution, as the observers will not see any notable differences. The use of H.265 codec is relevant only for the videos at the Ultra HD resolution, particularly at low bitrates. This is due to the fact that the quality of H.264 encoded video sequences increases with the rising bitrate up to the point where it reaches or even surpasses the perceived quality of H.265 sequences. Secondly, the compression efficiency of the H.265 compression standard at the Ultra HD resolution reaches the compression efficiency of both codecs at the Full HD resolution.

Indisputably, the conclusions drawn from the Analysis of Variance (ANOVA) and the graphical representation of the subjective quality evaluation results coincide. These findings could be beneficial for visual media content providers and broadcasting companies, as they indicate how to adjust video compression parameters to improve its quality. The fastest growth of perceived video quality is apparently due to an increase in bitrate. Specifically, the quality increases most rapidly until the bitrate reaches a value of approximately 5 Mbps. The analyses also revealed which combination of resolution and compression format is best used so that the resulting quality of visual content is perceived by viewers as good as possible.

### 4.5. Minimum Bitrate Thresholds Suggestions

Finally, Figure 10 shows the minimum bitrate thresholds at which the video sequences should be encoded to achieve good (4) or fair (3) quality. These quality thresholds are based on MOS values of used ACR method and are important for the bitrate setting of each codec to maintain a certain quality. Table 8 shows the mentioned minimum bitrates.

From Table 8, it follows that to achieve a good quality (value 4 on MOS scale), the video sequence must be coded to minimum 7.50 Mbps by both codecs for Full HD resolution and to 11.55 Mbps by H.264 codec and 9.00 Mbps by H.265 codec for Ultra HD resolution. To reach fair quality (value 3 on MOS scale), the minimum thresholds for the bitrates are 2.80 Mbps by H.264 codec and 2.60 Mbps by H.265 codec for Full HD resolution and 4.50 Mbps by H.264 codec and 2.80 by H.265 codec for Ultra HD resolution.

## 5. Conclusions

This paper dealt with the content impact on the perceived video quality evaluated using the subjective Absolute Category Rating (ACR) method. Eight types of video sequences with various scene content were evaluated. Two widely used video compression standards H.264/AVC and H.265/HEVC in combination with Full HD and Ultra HD resolutions, were tested. In the coding process, we selected 5 various bitrates based on our previous research, which showed that the efficiency of codecs grows nonlinearly with increasing bitrate. The number of bitrates was a compromise between the complexity and time requirements of subjective testing. In total, we created an annotated database which contains 160 different video sequences coded at constant bitrates with GOP set to half of the framerate value which is typical for video intended for transfer over a noisy communication channel. The perceived quality of the sequences was evaluated employing the subjective ACR method. The assessment was conducted in two laboratories: one situated at the University of Zilina, and the second at the VSB—Technical University in Ostrava. First, we calculated the correlation of the MOS values between both laboratories using the Pearson correlation coefficient (PCC) and the Root Mean Square Error (RMSE). The correlation proved to be considerably high. After that, we described the impact of the bitrate on video quality depending on scene content defined by Spatial (SI) and Temporal information (TI). The results showed that even if the sequences with low SI and TI values reach better MOS than the sequences with higher SI and TI values, these two parameters are not sufficient for scene description, and this domain should be the subject of further research. Subsequently, we described the impact of bitrate on video quality depending on codec and resolution. Based on the results, we concluded that the employment of the H.265 codec for compression of Full HD sequences is inessential, as the people did not observe any significant differences. Furthermore, we stated that the compression efficiency of the H.265 codec by the Ultra HD resolution reaches the compression efficiency of both codecs by the Full HD resolution. We also applied the ANOVA to verify what stemmed from the graphical representation of the subjective evaluation results. Finally, we determined the minimum bitrate thresholds at which the video sequences at both resolutions retain good and fair subjectively perceived quality.

## Figures and Tables

**Figure 1 sensors-21-02872-f001:**
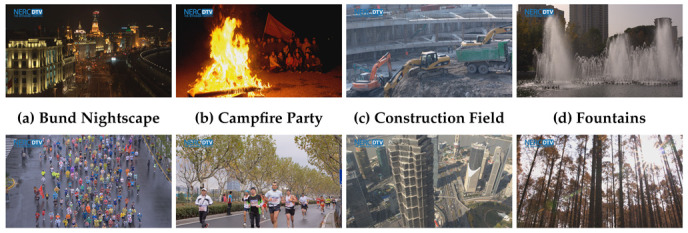
Printscreens of used test sequences. Reprinted with permission from [60], Copyright 2021, Uhrina.

**Figure 2 sensors-21-02872-f002:**
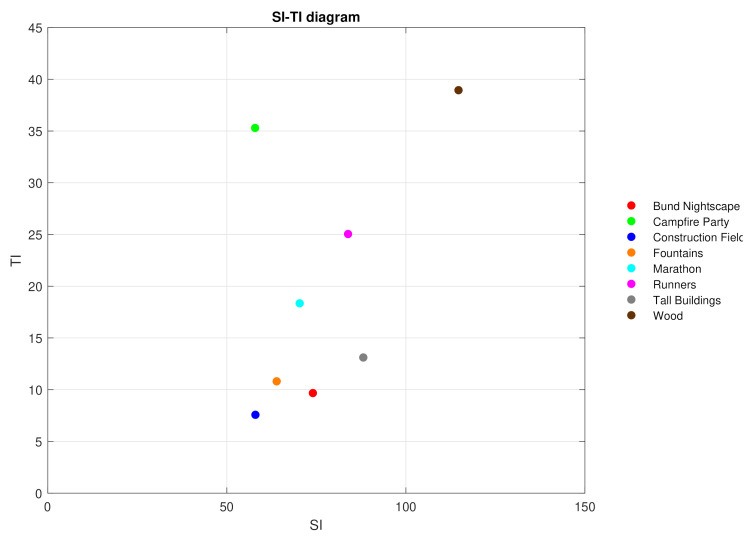
Spatial Information (SI) and Temporal Information (TI) diagram of used test sequences. Reprinted with permission from [60], Copyright 2021, Uhrina.

**Figure 3 sensors-21-02872-f003:**
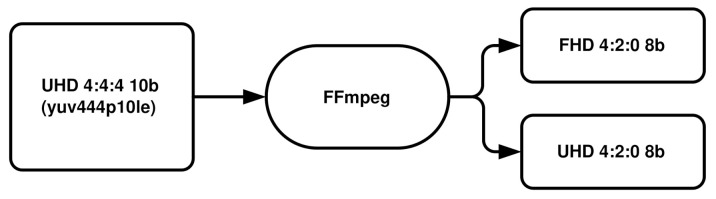
Process of preparing the test sequences: chroma subsampling, bit depth, and resolution changing.

**Figure 4 sensors-21-02872-f004:**
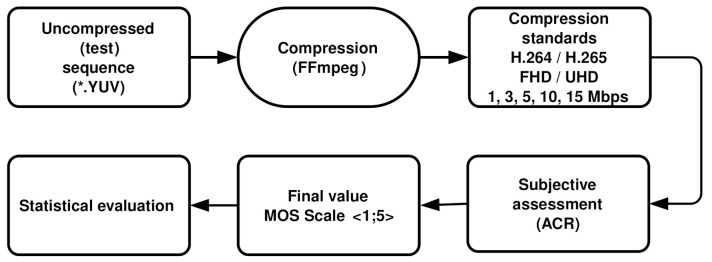
Complete process of coding and assessing the video quality.

**Figure 5 sensors-21-02872-f005:**
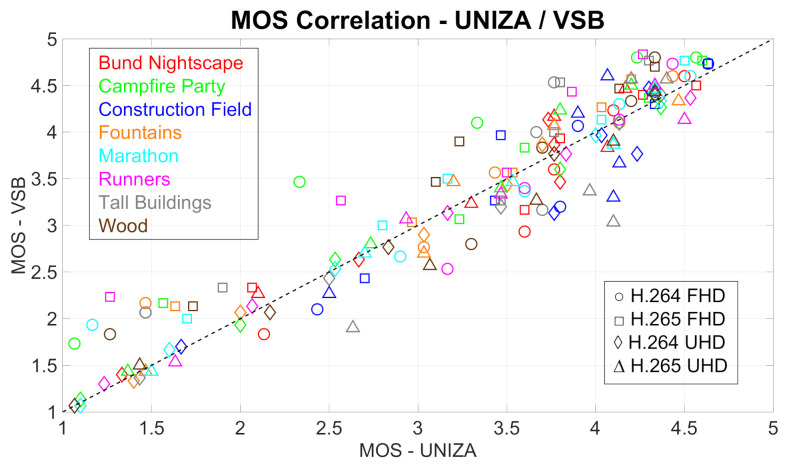
Comparison of Mean Opinion Score (MOS) values obtained from different laboratories. Each spot represents MOS values for corresponding codec, resolution, and test sequence

**Figure 6 sensors-21-02872-f006:**
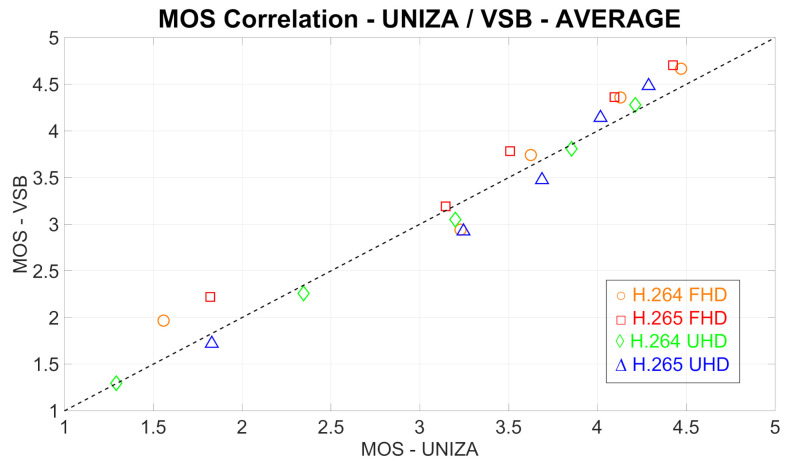
Comparison of MOS values obtained from different laboratories. Each spot represents averaged MOS values from particular test sequences for corresponding codec and resolution.

**Figure 7 sensors-21-02872-f007:**
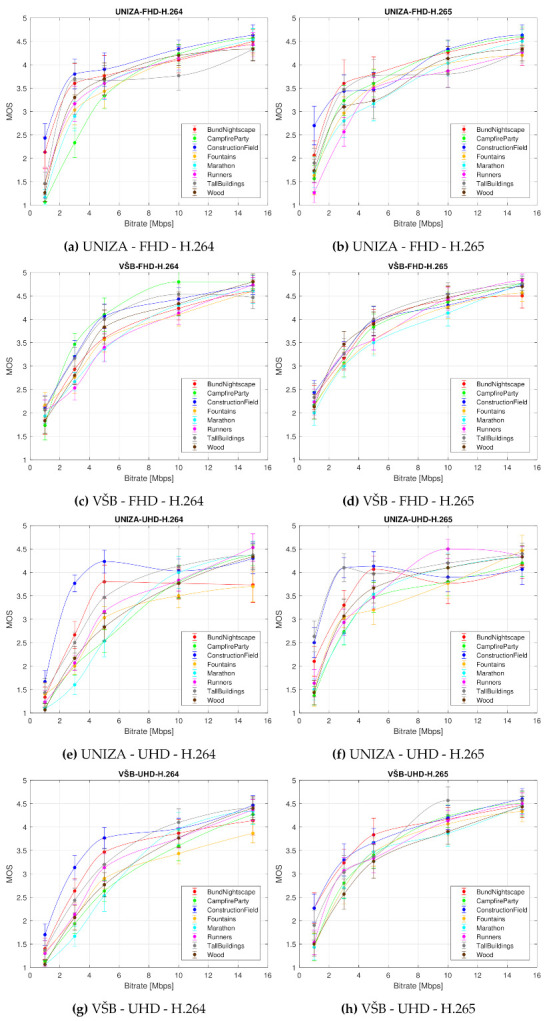
Bitrate impact on the perceived video quality (defined by the MOS score with associated Confidence Interval (CI)) depending on codec and resolution for both laboratories independently. Each curve represents MOS values for each type of used test sequence.

**Figure 8 sensors-21-02872-f008:**
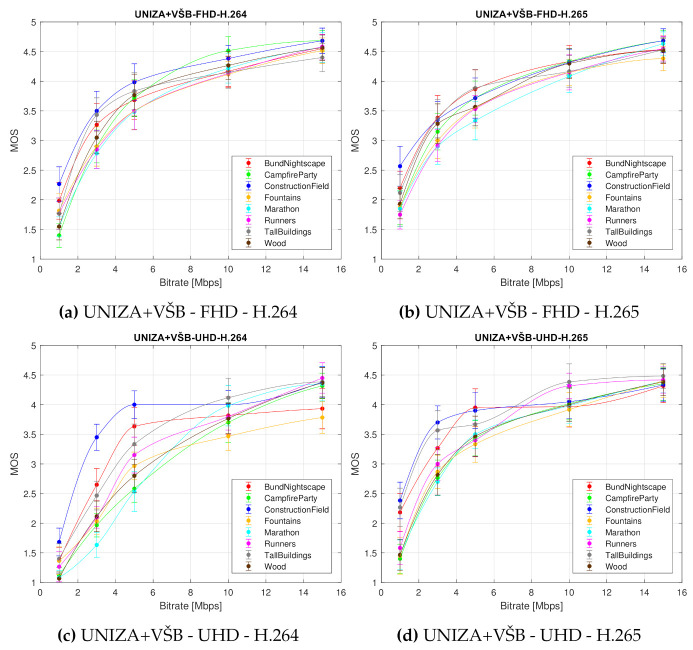
Bitrate impact on the perceived video quality (defined by the MOS score with associated CI) depending on the codec and resolution for both laboratories jointly. Each curve represents averaged MOS values from both laboratories for each type of used test sequence.

**Figure 9 sensors-21-02872-f009:**
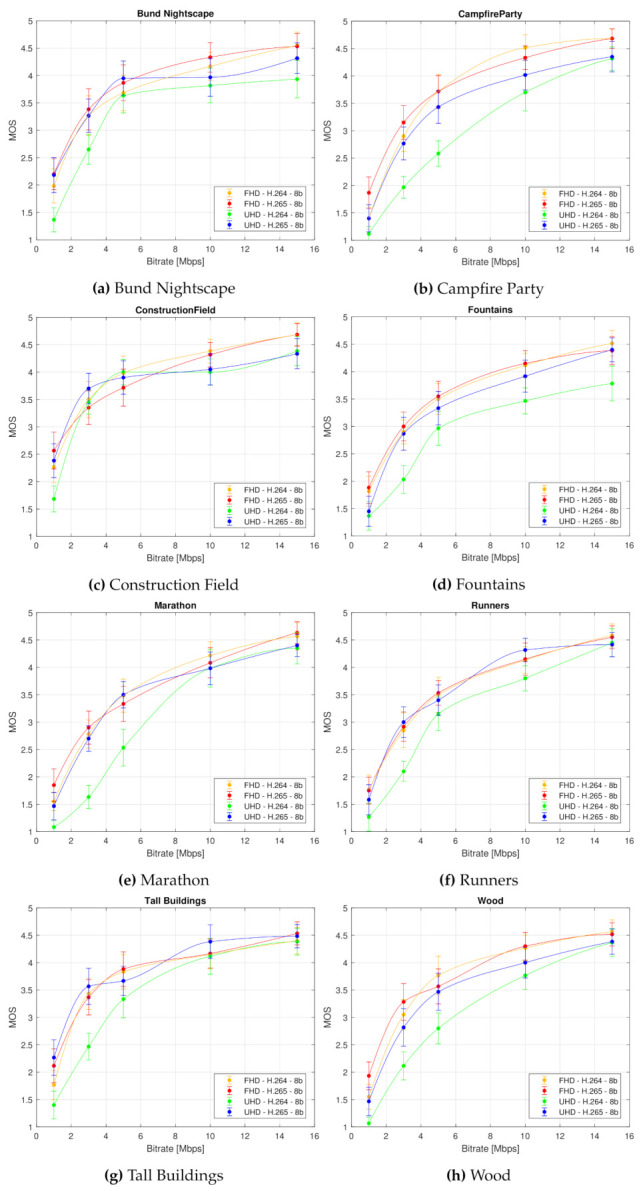
Bitrate impact on the perceived video quality (defined by the MOS score with associated CI) depending on used test sequence. Each curve represents averaged MOS values from both laboratories for corresponding codec and resolution.

**Figure 10 sensors-21-02872-f010:**
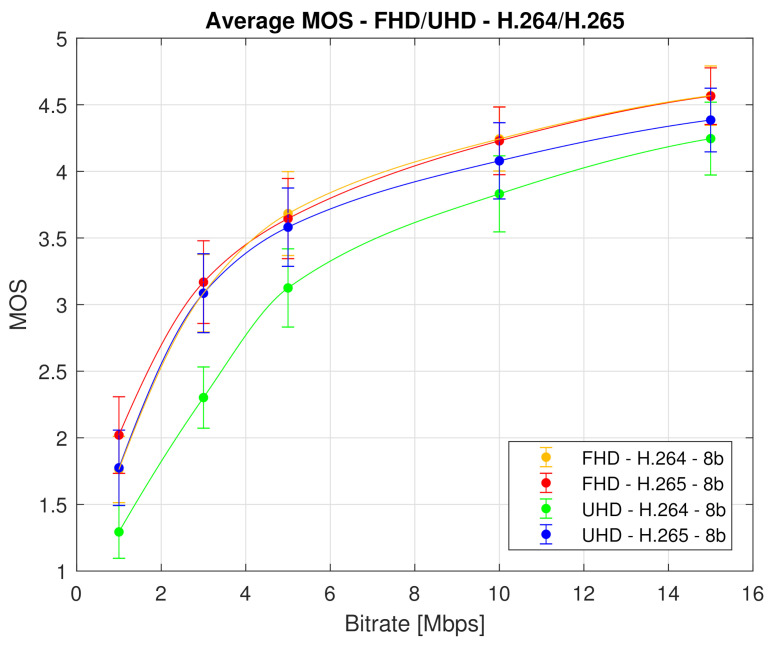
Bitrate impact on the perceived video quality (defined by the MOS score with associated CI). Each curve represents averaged MOS values from both laboratories for corresponding codec and resolution—average MOS score.

**Table 1 sensors-21-02872-t001:** Parameters of test sequences.

Resolution	Chroma Subsampling	Bit Depth	Aspect Ratio	Framerate [fps]	Length [Seconds]
3840 × 2160 (UHD)	4:4:4	10 bits per channel	16:9	30	10

**Table 2 sensors-21-02872-t002:** Characteristics of the test sequences.

Test Sequence	Description	Test Sequence	Description
Bund Nightscape	is a video sequence portraying the above view of a night city crossed by a busy road next the river. The time-lapse video is captured from a high angle with a steady camera as one extreme long shot. The scene is relatively static, except for the accelerated movement of cars driving on the road, people passing by, flags waving in the wind and flashing lights.	Marathon	is a video sequence picturing a large group of people in colorful apparel running a race on an asphalt road on a rainy day. The sequence was filmed from bird’s eye perspective with almost no camera movement as a very long shot. The scene is rather dynamic, given almost the entire frame is filled by running marathon participants and raindrops falling on the wet road.
Campfire Party	is a night time video sequence depicting a group of people posing for a photograph behind a large campfire. The long shot is captured by a stationary camera, which zooms in slightly at the end of the video. The motion in the scene is caused mainly by a flashing fire in the foreground and a woman who briefly runs out of and back into the shot.	Runners	is a video sequence that captures athletes running on a tree lined road in a cloudy weather. The racers in the very long shot are approaching the stationary camera, which is positioned approximately at their eye level. The scene contains a considerable amount of motion caused by rushing contestants and by the wind in the treetops.
Construction Field	is a very still video sequence capturing construction equipment in the middle of a building site during excavation work. A hand-held camera was used to film the very long shot from a high angle. The only moving objects in the scene are an excavator digging a foundation pit and people slowly walking in the background.	Tall Buildings	is a video sequence portraying the tallest skyscrapers and busy intersections in Shanghai, with a grand river in the background. The video was captured from a bird’s eye view using a camera that slowly pans to take a panoramic extreme long shot. The movement in the scene is primarily a result of the panning motion of the camera and partially of the cars driving fast at a deep distance.
Fountains	is a video sequence focused on several fountains in the left of a housing estate with multiple trees and apartment buildings in the background. The video is captured by a static camera as a long shot. All the motion in the scene can be attributed to water gushing from the fountain jets and droplets evaporating into the air.	Wood	is a video sequence picturing a tall forest during a sunny autumn day. The video was filmed from a low angle as a long shot with a camera performing a moderately fast panning motion. All the movement in the scene can be attributed to the camera pan and the resulting change in the angle of the sunlight rays incident on the lens.

**Table 3 sensors-21-02872-t003:** Types of used displays.

Type of Assessment	Type of Display
UNIZA − FHD	Samsung LE40C750R2W FHD
UNIZA − UHD	Samsung U24E590D UHD
VSB − FHD + UHD	24” Dell P2415Q UHD

**Table 4 sensors-21-02872-t004:** Statistical characteristic of the observers.

University	Resolution	Number of Men	Number of Women	Average Age
UNIZA	FHD	25	5	24
UNIZA	UHD	21	9	22
VSB	FHD + UHD	15	15	25
UNIZA + VSB	FHD + UHD	61	29	24

**Table 5 sensors-21-02872-t005:** Correlation of MOS score between the laboratories.

	**Pearson CC**	**RMSE**
**FHD-H.264**	0.97	0.30
**FHD-H.265**	0.99	0.31
**UHD-H.264**	1.00	0.10
**UHD-H.265**	0.98	0.23

**Table 6 sensors-21-02872-t006:** Three-way Analysis of Variance (ANOVA) using video codec as a criterion.

** H.264 **					
**Source of Variation**	**Sum of Squares**	**Degrees of Freedom**	**Mean Square**	***F*** **-Value**	***p*** **-Value**
Bitrate (X1)	2541.95	4	635.488	982.84	0
Scene Type (X2)	134.03	7	19.148	29.61	0
Resolution (X3)	106.68	1	106.682	164.99	0
X1*X2	106.58	28	3.802	5.89	0
X1*X3	34.24	4	8.561	13.24	0
X2*X3	12.16	7	1.737	2.69	0.009
Error	1518.18	2348	0.647		
Total	4453.83	2399			
** H.265 **					
Bitrate (X1)	1875.05	4	468.764	669.56	0
Scene Type (X2)	90.96	7	12.994	18.56	0
Resolution (X3)	0.12	1	0.12	0.17	0.6784
X1*X2	88.31	28	3.154	4.51	0
X1*X3	7.96	4	1.99	2.84	0.0229
X2*X3	30.65	7	4.379	6.25	0
Error	1643.85	2348			
Total	3736.91	2399			

**Table 7 sensors-21-02872-t007:** Three-way ANOVA using video resolution as a criterion.

**Full HD**					
**Source of Variation**	**Sum of Squares**	**Degrees of Freedom**	**Mean Square**	***F*** **-Value**	***p*** **-Value**
Bitrate (X1)	2210.04	4	552.509	806.21	0
Scene Type (X2)	82.43	7	11.776	17.18	0
Compression Standard (X3)	0.01	1	0.007	0.01	0.9214
X1*X2	79.43	28	2.837	4.14	0
X1*X3	11.11	4	2.779	4.05	0.0028
X2*X3	16.25	7	2.322	3.39	0.0013
Error	1609.13	2348	0.685		
Total	4008.4	2399			
**Ultra HD**					
Bitrate (X1)	2186.06	4	546.515	842.85	0
Scene Type (X2)	156.8	7	22.4	34.55	0
Compression Standard (X3)	112.23	1	112.234	173.09	0
X1*X2	145.9	28	5.211	8.04	0
X1*X3	52	4	12.999	20.05	0
X2*X3	12.31	7	1.759	2.71	0.0084
Error	1522.48	2348	0.648		
Total	4187.78	2399			

**Table 8 sensors-21-02872-t008:** Minimum bitrate thresholds to achieve good (4) and fair (3) video quality.

**MOS Scale**	**FHD-8b**		**UHD-8b**	
	**H.264**	**H.265**	**H.264**	**H.265**
Good (4)	7.50 Mbps	7.50 Mbps	11.55 Mbps	9.00 Mbps
Fair (3)	2.80 Mbps	2.60 Mbps	4.50 Mbps	2.80 Mbps

## Data Availability

The data presented in this study are available on request from the corresponding author.

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
