# Peer review of "Impact of Scene Content on High Resolution Video Quality"

_sensors, 2021, doi:10.3390/s21082872_

Round 1
Reviewer 1 Report
The paper assesses the impact of content on perceived video quality using the subjective absolute category classification (ACR) method. Eight types of video streams from the SJTU data set have been used, encoded by the H.264 / AVC and H.265 / HEVC video compression standards in Full HD and Ultra HD resolutions.
The article presents interesting conclusions about the resolution and the need to use one or the other codec, which is interesting for application purposes, and the minimum bitrate thresholds in which the videos must be encoded to achieve a good and fair quality.
However, authors should consider the following:
- Fig. 5: Problems with the graphic representation that must be solved.
- Figure 6 appears after Figure 7 in the text.
- Tables 6 and 7 show a three-way analysis of variance using the video codec as a criterion in figure 6 and the video resolution in table 7. However, the information presented in both tables is confusing because there is no reference to the video codec in the table. 6.
- Better structuring of the role should be provided. Section 4 is excessively long and should be divided into subsections for a better organization of the content.
Author Response
Dear reviewer,
thank you for your comments and suggestions, please find our responses attached.
Best regard

Reviewer 2 Report
The paper is well founded and explained. My suggestion is only to improve the reading of the text. 1. In "Introduction" section: I suggest changing the information of the paper by the name of the author. For example, in the text, the authors begin the reference with the term "In paper [reference]". I suggest putting, In [reference] or "The {name of authors} et al. [reference]". This eliminates repetition.2. Line 121, I suggest specifying the meaning of GOP, i.e., changing GOP by Group of Pictures (GOP). 3. Table 6, I suggest replacing X1, X2 and X3 by the names of the sources of variations.
4. In figures 7, 8 and 9 I suggest adding the label in each graphic (a) xxx, (b) xxx, etc. Thus, it is possible to identify in each graph what is being analyzed more clearly.
Author Response
Dear reviewer,
thank you for your comments and suggestions,
please find our responses attached.
Best regards

Reviewer 3 Report
Line 8: the abbreviations SI, TI should be explained
Line 11: is ‘sequences. and' should be ‘sequences and’
Line 57: is ‘method . In’ should be ‘method. In’
Line 87: measures TI SI should be shortly explained, only the reference is insufficient
Figure 1: remove path/filenames of the image files, each image is described by the letter a), b),… filenames may be specified in the figure caption
Figure 2: remove path/filename
Page 4: remove empty space below the Figure 2
Line 94: ‘unsampled (YUV 4:4:4)’ is not correct, digital images in any color format is sampled by definition. I propose to replace it by ‘YUV 4:4:4 color format’
Line 124: please explain how t he persons involved in the subjective tests were trained, did they have any experience in image/video processing
Figure 5: remove filenames, plot b) is moved to the right and partially invisible
Line 175: please add a reference for the ANOVA
Author Response

(The authors gave the same response as above.)

Reviewer 4 Report
This paper presents comparison between H.264/AVC and H.265/HEVC compression standards, at full HD and ultra HD resolutions. Results have been compared in 2 independent laboratories, showing high correlation between MOS scores. Results in the paper show that H.264/AVC and H.265/HEVC compression have similar MOS scores for full HD resolution, contrary to the many similar experiments that can be found in the literature, e.g. [1] with 50% bit-rate savings (based on MOS) for full HD, [2] with 61% of MOS BD-rate savings for full HD, [3] with average bitrate saving of 39% (using PSNR), etc.
There are several problems with the current setup that probably made the results as presented:
1. Section 2.3 should be explained with more details regarding the coder setup. Ffmpeg can be used with several presets that optimise speed vs quality, for the same final bitrate. E.g preset "placebo" is the slowest, with the best quality, while preset "ultrafast" is very fast but with lower quality.
2. It is not clear how did you choose bitrate? Have you used a constant or average bitrate option? E.g. probably the best setup would be (to have the best final quality with the specified bitrate, to be able to compare different codecs) 2-pass average bitrate. CRF option might be also used, but in this case you don't know in advance what is the final bitrate.
3. It is not clear if the option to use the same GOP should be used. In general case, both H.264/AVC and H.265/HEVC should be used with longer GOP structure (in the paper it is 15 frames, why?), to be able to use developed tools for better coding efficiency. This might also have some impact on the presented results.
4. Finally, to be able to fairly compare both standards, reference codecs should be used: HM for H.265/HEVC and JM for H.264/AVC. However, ffmpeg and other tools will probably produce similar results if they are properly configured.
5. It is interesting to compare compression efficiency, related video quality, and how it depends on different parameters such as SI and TI.
6. Table 6, probably it should be H.264/AVC instead of Full HD and H.265/HEVC instead of Ultra HD?
[1] J.-R. Ohm; G. J. Sullivan; H. Schwarz; T. K. Tan; T. Wiegand (December 2012). "Comparison of the Coding Efficiency of Video Coding Standards – Including High Efficiency Video Coding (HEVC)" (PDF). IEEE Transactions on Circuits and Systems for Video Technology. IEEE. 22 (12).
[2] T. K. Tan et al., "Video Quality Evaluation Methodology and Verification Testing of HEVC Compression Performance," in IEEE Transactions on Circuits and Systems for Video Technology, vol. 26, no. 1, pp. 76-90, Jan. 2016, doi: 10.1109/TCSVT.2015.2477916.
[3] D. Grois, D. Marpe, A. Mulayoff, B. Itzhaky and O. Hadar, "Performance comparison of H.265/MPEG-HEVC, VP9, and H.264/MPEG-AVC encoders," 2013 Picture Coding Symposium (PCS), San Jose, CA, USA, 2013, pp. 394-397, doi: 10.1109/PCS.2013.6737766.
Author Response

(The authors gave the same response as above.)

Round 2
Reviewer 4 Report
Authors have answered the questions in the previous review. Still, I would recommend the following:
In abstract: "The evaluation results led us to the conclusion that it is unnecessary to use the H.265/HEVC codec for compression of Full HD sequences and the compression efficiency of the H.265 codec by the Ultra HD resolution reaches the compression efficiency of both codecs by the Full HD resolution." You might add the information somewhere - using the constant bitrate and GOP of 15 frames, which is "typical for video intended for transfer over a noisy communication channel" (This is now mentioned in section 2.3). Same might be added in the conclusion.
Author Response

(The authors gave the same response as above.)
